# Seasonality of Deaths Due to Heart Diseases among Cancer Patients

**DOI:** 10.3390/medicina58111651

**Published:** 2022-11-15

**Authors:** Velizar Shivarov, Hristo Shivarov, Angel Yordanov

**Affiliations:** 1Department of Experimental Research, Medical University Pleven, 5800 Pleven, Bulgaria; 2Singing River Hospital, Pascagoula, MS 39567, USA; 3Department of Gynecologic Oncology, Medical University Pleven, 5800 Pleven, Bulgaria

**Keywords:** oncocardiology, death, cancer, season, cosinor model, SEER program

## Abstract

*Background and Objectives*: Cancer patients are at increased short- and long-term risk of cardiac toxicity and mortality. It is well-known that cardiac morbidity and mortality follows a seasonal pattern. Here we address the question of whether heart disease-related fatalities among cancer patients also follow a seasonal pattern. *Materials and Methods*: We performed a retrospective analysis of seasonality of deaths due to heart diseases (*n* = 503,243) in patients with newly diagnosed cancer reported during the period from 1975 to 2016 in the US’s largest cancer registry—the Surveillance, Epidemiology, and End Results (SEER) database. Seasonality was assessed through a classical cosinor model assuming a single annual peak. *Results*: We identified a significant seasonal peak in the first half of November. A peak with identical features was for all subgroups of patients defined based on demographic characteristics. This was also the case when analysis was performed on subgroups defined by the type of malignancy. Only patients with acute leukemias, pancreatic cancer and nervous system malignancies did not have a seasonal pattern in heart disease-related fatalities. *Conclusion*: the rate of heart disease-related fatalities after cancer diagnosis follows a seasonal pattern similar to that observed for the general population, albeit with an earlier peak in November. This suggests that close monitoring of the cardiovascular system in cancer survivors must be particularly active from late autumn and during the entire winter period.

## 1. Background

Heart diseases and cancer are the first- and the second-most common cause of death in the USA, respectively [1]. It is a well-documented phenomenon that cancer patients have an increased risk of cardiovascular morbidity and mortality [2,3]. The major contributing factors to the increased risk for cardiovascular diseases in cancer patients are the cardiac and vascular toxicity of chemotherapy, radiotherapy and even targeted therapy [4,5]. It is well-known that cardiovascular diseases show a seasonal variation in incidence and mortality, with a usual peak in the cold months of the year due to the complex interplay between physiological and environmental factors [6,7]. Indeed, many physiological parameters such as blood pressure, heart rate, cholesterol level, body weight, inflammatory and coagulation parameters show seasonal variation [8,9,10,11]. Most of these are believed to be a direct consequence of the seasonal variation in environmental factors such as daylight duration, temperature or humidity [7,9]. Besides, environmental factors can affect physiological parameters indirectly through seasonal variations in diet and physical activity [12,13]. To the best of our knowledge, there is no real-world study that evaluated the putative modulatory effect of cancer diagnosis on seasonality of cardiovascular morbidity and mortality. This question is of obvious biological and clinical interest. From a biological and pathological standpoint, it is important to understand whether cancer-associated risk for cardiovascular diseases affects the general seasonal pattern of cardiovascular diseases, or it remains similar to the one for the general population. From a clinical standpoint, the precise understanding of the timing of possible peaks in cardiovascular morbidity and mortality in patients with cancer can determine the risk-based management of those patients, and eventually improve the clinical outcomes. Therefore, here we questioned whether the rate of deaths due to heart disease in cancer patients followed a seasonal pattern similar to the one observed for the general population.

## 2. Materials and Methods

### 2.1. Data Availability

For the purposes of this study, it was possible to extract records of cancer patients with reported cause of death from the US’s largest cancer registry (Surveillance, Epidemiology, and End Results (SEER) program). All patients were included in this study were reported in the SEER database during the period from 1975 to 2016 (November 2018 submission), as already described previously [14]. The total number of extracted unique reports of patients with cause of death recoded to “Diseases of Heart” was 503,243. Notably, this set of cases included only the entries of the first diagnosed primary malignancy. The month of death of each patient is not provided in the SEER database. Therefore, we estimated it for all patients based on the reported month and year of diagnosis and survival time in months. For the purposes of that estimation, we accepted that the average duration of any month is 30.25 days. In order to classify counties into Southern and Northern ones, we obtained their geospatial locations from the website of the US Census bureau https://www.census.gov/geographies/reference-files/time-series/geo/gazetteer-files.html, accessed on 30 March 2022, as described previously [14]. The median value of the latitudes of the counties of residence for all patients included in this analysis was 37°56’22″ N; this median value was used to classify a county as a Northern or Southern one, as described previously [14]. We also classified the counties into metropolitan and nonmetropolitan ones based on their rural-urban classification in 2003 (https://seer.cancer.gov/seerstat/variables/countyattribs/ruralurban.html, accessed on 30 March 2022). Participants’ written informed consent was not required, as this study used only publicly available anonymized data from a cancer registry. Per local regulations, this study was considered exempt from an ethics committee review, as it was based on publicly available cancer registry data.

### 2.2. Statistical Analysis

Previous studies showed that mortality from cardiovascular diseases follows a sinusoidal pattern, with a single peak during the year [13,15,16]. The most suitable approach to model this seasonal variation in cardiovascular deaths was the classical cosinor model [13]. Furthermore, we successfully implemented the same model to analyze seasonality of suicides among US cancer patients [14]. Therefore, we interrogated the presence of circannual pattern in deaths from heart diseases using a cosinor model, with one cycle per year [14,16]. The general formula of the model is:$$f\left(t\right)=Acos\left(\frac{2\pi t}{c}-P\right)$$

In this formula, *A* denotes the amplitude of the sinusoidal curve, whereas *P* denotes its phase and *c* the duration of the seasonal cycle (*c* = 12 for 1 cycle per year), and *t* denotes the time of each observation [13,14]. This can be linearized to the following equation:$$Y_{t}=c\cos\left(\omega_{t}\right)+s\sin\left(\omega_{t}\right),\;t\;=\;1,\;\ldots ,\;n.$$

Amplitude and the phase can be derived using the estimates *c* and *s* from the equation above [13,14]. For the seasonality statistical tests, the level of significance for the cosine and sine terms was set to less than 0.025, in order to keep the overall significance level of the model at α = 0.05 [13,14]. Data processing and statistical analyses were performed on R environment for statistical computing v. 4.2.0 for Windows (64-bit), with the use of the package *season* (v. 0.3.15) [17]. All figures for publication were produced with the R package *ggpubr* (v. 0.4.0). 

## 3. Results

The SEER database is an invaluable source for real-world data for analysis of outcomes in cancer patients. Using this, we managed to extract a total of 503,243 records of primary malignancies and reported terms for the cause of death “Diseases of Heart” reported during the period from 1975 and 2016. Demographic features of the patients included in the current analysis are represented in Table 1, and expectedly represented the overall distribution known from previous reports, which focused on descriptive analysis of cardiovascular deaths in US cancer patients [2,3]. In addition to the data directly reported in the SEER database for each patient, we derived the date of death and the age at death by addition of the available survival duration to the month of diagnosis, as described above. The median of the estimated age at death was 82 years, with a range between 0 and 118 years (Table 1).

We fitted a classical cosinor model with one cycle per year to the cohort of all patients included with death due to heart disease. We identified a significant seasonal peak during the first half of November (Table 2 and Figure 1). Notably, all subgroups defined based on demographic features such as gender, age at death, race, time of death after initial cancer diagnosis, geographic location of the county of residence and period of diagnosis showed significant seasonal peaks. Besides, the peaks were almost invariably identified in the first half of November, with the exception of Black patients and patients who died within the first year of diagnosis (early death). Black patients showed a slightly earlier peak in late October, while for patients with death within the first year of diagnosis the peak was shifted to early December (Table 2 and Figure 1). Patients diagnosed between 2010 and 2016 showed an earlier peak, which could be attributed to a shorter follow-up for patients diagnosed in the second half of 2016 (Table 2 and Figure 1). 

The dataset included cancer patients with different entities at different stages requiring different types of therapy. It was rational to accept that not all malignancies would have seasonal variation in mortality from heart diseases. To address this question, all patients were regrouped by entity in 16 major categories (acute leukemias, chronic leukemias, lymphomas and myelomas, breast cancer, lung and bronchial cancer, colorectal cancer, pancreatic cancer, other gastrointestinal (GI) malignancies, head and neck cancer, endocrine malignancies, female genital malignancies, male genital malignancies, prostate cancer, urinary system malignancies, nervous system malignancies, melanoma of the skin, soft tissues and bone cancers and miscellaneous malignancies), as we did in our previous study [14]. Analogous to the analyses described above, we fitted a cosinor model with a single cycle per year to each of these diagnosis-defined subgroups. Those analyses showed significant seasonality in heart-related deaths at an alpha level of 0.025 for all subgroups, with only three exceptions: patients with acute leukemias; pancreatic cancer; and nervous system malignancies (Table 2, Figure 2).

## 4. Discussion

Cardiovascular diseases (ischemic heart disease and stroke) are a major global health issue [18]. It has been clearly documented that incidence, mortality and hospitalizations due to cardiovascular diseases and heart failure follow a clear seasonal pattern over a range of geographic locations [6,19,20,21]. The almost universal peak in cold months may simply follow the dynamics of cardiovascular factors, and be further modified by a plethora of physiological (e.g., decreased vitamin D levels, increased platelet activation) and environmental (e.g., dietary intake, physical activity) factors [19].

Cancer is the second-most common cause of death in developed countries, including the US [1,22], and cancer patients and survivors are at increased risk of heart disease-related death [2]. On the other hand, cancer-related mortality is less amenable to seasonal variations [6,22]. To the best of our knowledge, however, the question of whether cancer diagnosis usually associated with multimodal therapy that may have cardiac and vascular toxicity modifies the seasonal pattern of heart disease-related mortality has not been addressed by the scientific literature. This question is of obvious biological and clinical importance, as long-term cancer survivors (e.g., Hodgkin lymphoma patients) suffer from increased long-term mortality due to cardiac diseases as a consequence of the cardiotoxic chemo- and radiotherapy at younger ages. Understanding of the interplay between cancer and seasonality of cardiovascular diseases can help in dissecting the pathophysiological pathways of cardiovascular diseases in cancer patients, and eventually tailoring their management over time in order to avoid excessive cardiovascular mortality.

Here we addressed this question using data from 503,243 unique cases of heart disease-related fatalities after first cancer diagnosis in patients from all ages diagnosed and reported in the SEER database between 1975 and 2016 (Table 1). Over the last few decades, there have been several methods developed to model seasonal health data, which can be classified into three main groups: comparison of discrete time periods; geometrical models; and generalized linear models [13,23]. The most popular approach is the standard cosinor model, due to its simplicity and easy interpretation [13,14]. The model assumes a sinusoidal pattern of the rate of events over the time period, and allows estimation of the phase and amplitude of regular or irregular time series following a sinusoidal pattern [13]. The cosinor model has been widely used to model seasonal health data, including laboratory parameters [9,11,24], incidence of infectious diseases [25], and cancer incidence and death [14,26]. A number of studies used the cosinor model to demonstrate that cardiovascular morbidity and mortality follows a circannual pattern with a single peak in the winter [15,16,27]. Therefore, we also tested that model on the numbers of heart fatalities per month among all patients and various subgroups of cancer patients defined either by demographic characteristics or by the primary entity. In the entire cancer patients group, we observed a significant peak in the number of heart-related deaths in early November. Notably, this late autumn peak is different from the peaks reported for deaths from all causes and from cardiovascular diseases in the US, which are during the winter period (January and February) [22]. However, cardiovascular mortality in the US also has a single circannual peak [22], and this further justifies our approach to use a cosinor model with a single peak during the year. As we estimated the date of deaths based on month of diagnosis and survival of months, we may have an inherent bias towards identifying an earlier peak, but the maximum lag of the actual peak would be no more than one month later (i.e., early December). In any case, it is obvious that the peak in cardiac mortality in cancer patients is earlier than that of the general population. This may be a true phenomenon with a clear mechanistic explanation, as the cardiovascular system of cancer patients may be more sensitive to the short-term effect of abrupt change in some environmental factors, such as ambient temperature [28,29,30,31]. Notably, the seasonality of cardiac deaths was also observed during the first year after diagnosis, but the peak was in early December, which suggests that indeed the earlier peak in November for long-term survivors might be due to the altered sensitivity of the cardiovascular system to seasonal factors after prolonged exposure to multimodal anticancer therapy. Furthermore, this phenomenon might be even more pronounced in patients with predisposing genetic polymorphisms for cardiac toxicity from specific commonly used cytotoxic drugs such as anthracyclines [32]. Besides, this suggests that cancer patients with more aggressive diseases, and eventually more advanced stages who have a shorter overall survival, also have a seasonal variation in cardiac mortality with a single peak. We did not test seasonality in patients with different stages, as staging systems changed over time, and are inconsistently reported in the SEER database, making such analyses over long periods of time unreliable. Finally, the seasonal pattern in heart diseases-related deaths was preserved in almost all subgroups of patients defined by cancer subtype. The three exceptions—patients with acute leukemia, nervous system cancer and pancreatic cancer—suffer from more aggressive diseases with shorter overall survival, in which mortality is usually driven by the underlying primary malignancy and not by other chronic diseases.

Unfortunately, the current study suffers from several limitations that are a direct consequence of the used source database. The SEER database does not provide the actual date of death or the specific cause of death. To overcome some of these limitations, we estimated the month and age of death based on the provided month of diagnosis and survival duration. Unfortunately, we could not assess the seasonality of specific heart diseases, i.e., cardiovascular (including acute myocardial infarction) vs. heart failure. Furthermore, the SEER database does not provide specific information regarding the type of chemotherapy, radiotherapy and targeted therapy used in any case, so that their effect on the seasonality of cardiac disease-related deaths could not be evaluated. However, this registry data provide an extremely large number of cardiac disease-related deaths documented in an unbiased fashion over five decades, which makes our approach and assessment robust.

## 5. Conclusions

In sum, our findings suggest that the rate of heart disease-related fatalities after cancer diagnosis follows a seasonality similar to that observed for the general population. An analogous pattern is identified in all subgroups of patients defined by demographic characteristics, and in most subgroups defined by entity. The incidence of heart disease-related deaths peaked earlier than in the general population, which may be due to alteration in the cardiovascular system associated with cancer therapy, making it more sensitive to the first changes in bioclimatic factors in late Autumn. This hypothesis requires further in-depth epidemiological and experimental validation. However, our findings have important practical implications in the field of oncocardiology; they suggest that close monitoring of the cardiovascular system, including testing of serum biomarkers and prophylactic use of antiplatelet drugs in cancer survivors, must be particularly active from late autumn and during the entire winter period, as this could potentially prevent a significant number of deaths [33].

## Figures and Tables

**Figure 1 medicina-58-01651-f001:**
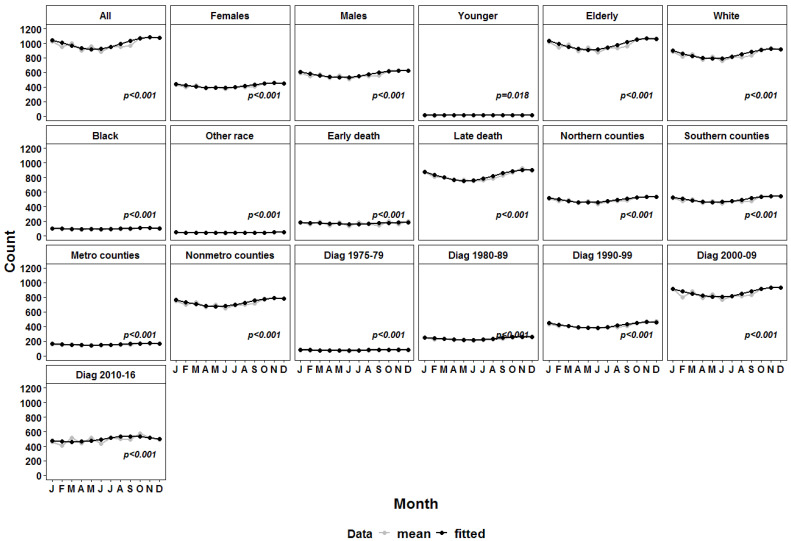
Sinusoidal curves of the number of deaths due to diseases of the heart in all patients and main subgroups (in grey) versus the fitted values (in black).

**Figure 2 medicina-58-01651-f002:**
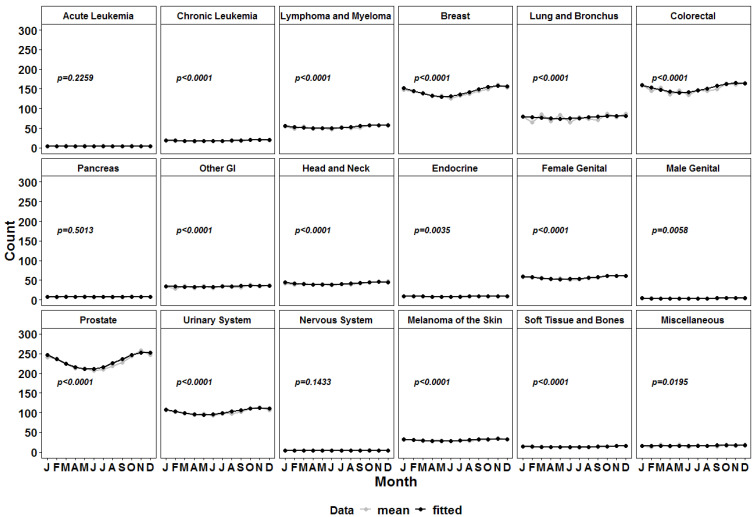
Sinusoidal curves of the number of deaths due to diseases of the heart per month in the main cancer subtypes (in grey) versus the fitted values (in black).

**Table 1 medicina-58-01651-t001:** Demographic characteristics of the patients included in the analysis.

	Numbers
Total	503,243
Sex	
Male	291,300
Female	211,943
Age at diagnosis	
Median	74
Range	(0–109)
Age at death	82
Range	(0–118)
Race	
White	11,109
Black	431,143
Other	21,116
Unknown	1035
Year of diagnosis	
1975–79	37,045
1980–89	103,862
1990–99	135,174
2000–09	176,583
2010–16	41,786

**Table 2 medicina-58-01651-t002:** Parameters of the cosinor model fits to all patients and defined subgroups.

Group	Count	Amplitude	Peak Month	Lowest Month	*p*-Value Cosine Term	*p*-Value Sine Term
All	503,243	83.56082394	11.2	5.2	9.40 × 10^−124^	3.33 × 10^−234^
Males	291,300	48.69524313	11.2	5.2	4.12 × 10^−72^	1.97 × 10^−139^
Females	211,943	34.86821062	11.2	5.2	1.55 × 10^−53^	7.66 × 10^−97^
Younger	5938	0.623275734	11	5	0.160744495	0.01778296
Elderly	497,305	82.91270153	11.2	5.2	1.32 × 10^−123^	6.17 × 10^−233^
White	431,143	72.70064142	11.2	5.2	1.02 × 10^−113^	1.14 × 10^−202^
Black	49,949	7.477840534	10.7	4.7	2.37 × 10^−5^	1.07 × 10^−26^
Other race	22,151	3.925627819	11.7	5.7	3.91 × 10^−12^	2.00 × 10^−8^
Northern counties	250,141	41.38915053	11.1	5.1	8.50 × 10^−58^	6.47 × 10^−121^
Southern counties	253,102	42.20022138	11.2	5.2	4.84 × 10^−68^	2.21 × 10^−115^
Early death	87,398	11.64307986	12.1	6.1	3.84 × 10^−33^	1.64 × 10^−10^
Late death	415,845	79.36887567	11.3	5.3	5.66 × 10^−144^	4.46 × 10^−238^
Metropolitan counties	77,570	13.74197	11.1	5.1	3.38 × 10^−21^	2.82 × 10^−44^
Nonmetropolitan counties	368,014	58.94964	11.2	5.2	5.22 × 10^−82^	1.58 × 10^−163^
Diagnosed 1975–79	37,045	7.915556	10.9	4.9	1.01 × 10^−10^	1.46 × 10^−32^
Diagnosed 1980–89	103,862	24.20887	11.3	5.3	3.45 × 10^−45^	1.82 × 10^−67^
Diagnosed 1990–99	135,174	42.73926	11.3	5.3	1.87 × 10^−59^	9.10 × 10^−84^
Diagnosed 2000–09	176,583	69.05977	11.5	5.5	1.62 × 10^−60^	1.14 × 10^−56^
Diagnosed 2010–16	41,786	40.26504	9.1	3.1	2.31 × 10^−7^	1.55 × 10^−23^
Acute Leukemia	2176	0.171431424	7	13	0.225941157	0.975550836
Chronic Leukemia	9264	1.858353742	10.9	4.9	0.002736572	6.77 × 10^−9^
Lymphoma and Myeloma	26,939	4.418684708	11.1	5.1	5.61 × 10^−7^	9.24 × 10^−15^
Breast	72,400	14.35817318	11.2	5.2	7.59 × 10^−28^	1.72 × 10^−47^
Lung and Bronchus	39,317	3.853408759	11.2	5.2	0.00010045	3.18 × 10^−8^
Colorectal	77,005	12.63774377	11.2	5.2	7.02 × 10^−20^	1.02 × 10^−36^
Pancreas	3813	0.122233124	3.6	9.6	0.881352942	0.5013316
Other GI	17,331	2.129560474	10.5	4.5	0.171230547	5.80 × 10^−08^
Head and Neck	21,166	3.695802261	11.2	5.2	2.77 × 10-^7^	2.94 × 10^−12^
Endocrine	3966	0.719667315	11.1	5.1	0.058285351	0.003497575
Female Genital	28,625	4.517477734	11.3	5.3	1.54 × 10^−8^	7.80 × 10^−13^
Male Genital	1707	0.480508962	11.2	5.2	0.053338149	0.005845772
Prostate	116,347	22.73642134	11.5	5.5	1.24 × 10^−56^	1.07 × 10^−58^
Urinary System	51,881	8.712249269	11.1	5.1	2.94 × 10^−12^	1.60 × 10^−28^
Nervous System	1850	0.201996926	10	4	0.9922066	0.143332111
Melanoma of the Skin	14,753	2.537777583	11.2	5.2	7.07 × 10^−5^	1.77 × 10^−8^
Soft Tissue and Bones	6529	1.647673764	11.8	5.8	1.48 × 10^−7^	0.000101968
Miscellaneous	8174	0.606991538	10.2	4.2	0.807253236	0.019465674

## Data Availability

Source data are publicly available through the Surveillance, Epidemiology and End Results (SEER) Program.

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
