# Peer review of "Seasonality of Deaths Due to Heart Diseases among Cancer Patients"

_medicina, 2022, doi:10.3390/medicina58111651_

Round 1

Reviewer 1 Report

I commend the authors on their clinically important and methodologically sound retrospective study of seasonality (with classical cosinor model) and cardiovascular mortality including by primary malignancy. I have no significant critiques.

Author Response

I commend the authors on their clinically important and methodologically sound retrospective study of seasonality (with classical cosinor model) and cardiovascular mortality including by primary malignancy. I have no significant critiques.

Reply: Thank you for the encouraging words. We made changes in accordance with the other reviewers’critique.

Reviewer 2 Report

To evaluate the influence of season on the heart disease-related death among participants with cancer, this author conducted a retrospective study. From this study, this author identified a significant seasonal peak in the first half of November. And only patients with acute leukemias, pancreatic cancer and nervous system malignancies did not have a seasonal pattern in heart diseases-related fatalities. From this study, this author suggests that close monitoring of the cardiovascular system in cancer survivors must be particularly active from late autumn and during the entire winter period. Aa reviewer, I thought this author focus on interesting topic. However, I also thought that to make present conclusion, this author should perform more careful analysis. Then I commented those concerns to this author.

1. In the first paragraph of abstract section, this author described as “Cancer patients are at increased short- and long-term risk of cardiac toxicity and mortality.” And next, this author described as “Here we addressed the question whether heart diseases-related fatalities among cancer patients also follow a seasonal pattern.” However, later sentences could not be supported by the former sentence. Therefore, this author should describe about the influence of season on heart diseases-related fatalities between those sentences.

2. This is a retrospective study that followed participants with newly diagnosed cancer. However, there is no information about the number of participants with newly diagnosed participants, number of deaths from heart related disease and the number of deaths from other disease. Information about those should be shown in abstract section.

3. Since this is a retrospective study, the length of follow up period also should be shown in abstract section.

4. Participants of this study is those who were newly diagnosed cancer reported during the period from 1975 to 2016. And the endpoint of this study is death from heart related disease. However, thanks to the advance in medical science and medical service, the death rate from cardiovascular disease might be reduced dramatically during those follow up period. Then this author should perform the analysis stratified by a business year. The length of follow-up period is too wide to ignore those factors.

5. From present study, this author concluded as “This suggests that close monitoring of the cardiovascular system in cancer survivors must be particularly active from late autumn and during the entire winter period.”. If monitoring of the cardiovascular system is efficient way to avoid the death from heart related disease, the death from heart related disease could be reduced during follow up period because of the advance in medical science and medical service. Is there any evidence that support this conclusion?

6. How is the influence of the length of period between diagnosed as having cancer and the death from heart related disease? From this study, this author found that only patients with acute leukemias, pancreatic cancer and nervous system malignancies did not have a seasonal pattern in heart diseases-related fatalities. However, those malignancies are known disease that is often diagnosed in advanced stage. Length of the life living with cancer should be taken into consideration.

7. Geographical issue also might influence on present results. Living in city and living in countryside should have quite different in accessibility for medical service. Therefore, difference between living in city area and living in country side area might strong influence on present results. If only country side area shows seasonal difference on present results, accessibility for medical service but not season itself might be an important determinant factor on present results.

8. Age also could influence on present results.

9. In material and method section this author described as “Notably, this et of cases included only the entries of the first diagnosed primary malignancy. The month of death of each patient is not provided in the SEER database. Therefore, we estimated it for all patients based on the reported month and year of diagnosis and survival time in months. For the purposes of that estimation we accepted that the average duration of any month is 30.25 days.” However, I could not understand the meaning of those sentences. What type of report that showed month and year of diagnosis and survival time? Is this means SEER database have the information of month and year of diagnosis and survival time?

10. Severity of malignant disease at the time of diagnosed also should be taken into consideration.

11. There are many cardiovascular risk factors that also regarded as risk factor for cancer. Therefore, many participants with cancer also might have a higher risk of cardiovascular risk factors. In this case, cardiovascular risk factor but not malignant disease determines the season of death from heart related disease. In this case, cancer itself never influence on the season of death from cardiovascular disease.

Author Response

To evaluate the influence of season on the heart disease-related death among participants with cancer, this author conducted a retrospective study. From this study, this author identified a significant seasonal peak in the first half of November. And only patients with acute leukemias, pancreatic cancer and nervous system malignancies did not have a seasonal pattern in heart diseases-related fatalities. From this study, this author suggests that close monitoring of the cardiovascular system in cancer survivors must be particularly active from late autumn and during the entire winter period. Aa reviewer, I thought this author focus on interesting topic. However, I also thought that to make present conclusion, this author should perform more careful analysis. Then I commented those concerns to this author.

  1. In the first paragraph of abstract section, this author described as “Cancer patients are at increased short- and long-term risk of cardiac toxicity and mortality.” And next, this author described as “Here we addressed the question whether heart diseases-related fatalities among cancer patients also follow a seasonal pattern.” However, later sentences could not be supported by the former sentence. Therefore, this author should describe about the influence of season on heart diseases-related fatalities between those sentences.

Reply: We added the following sentence: “It is well-known that cardiac morbidity and mortality follows a seasonal pattern.”

  1. This is a retrospective study that followed participants with newly diagnosed cancer. However, there is no information about the number of participants with newly diagnosed participants, number of deaths from heart related disease and the number of deaths from other disease. Information about those should be shown in abstract section.

Reply: This study included only death cases due to heart disease. We reported the total number of cases included in the abstract as requested by the reviewer. It is irrelevant to report counts of cases in the SEER database that were not included in the analysis and were not extracted at all.

  1. Since this is a retrospective study, the length of follow up period also should be shown in abstract section.

Reply: This study did not perform any time to event analysis (e.g. median overall survival). It only accounts for number of deaths reported in SEER between 1975 and 2016. Therefore no time of follow-up is to be reported in the abstract.

  1. Participants of this study is those who were newly diagnosed cancer reported during the period from 1975 to 2016. And the endpoint of this study is death from heart related disease. However, thanks to the advance in medical science and medical service, the death rate from cardiovascular disease might be reduced dramatically during those follow up period. Then this author should perform the analysis stratified by a business year. The length of follow-up period is too wide to ignore those factors.

Reply: We appreciate this recommendation. We agree and therefore we included a split of the cases per period of diagnosis. The data were presented in Table 2 and Figure 2.

  1. From present study, this author concluded as “This suggests that close monitoring of the cardiovascular system in cancer survivors must be particularly active from late autumn and during the entire winter period.”. If monitoring of the cardiovascular system is efficient way to avoid the death from heart related disease, the death from heart related disease could be reduced during follow up period because of the advance in medical science and medical service. Is there any evidence that support this conclusion?

Reply: There is no formal evidence. It is rational to accept that more active monitoring during periods of higher mortality rates can mitigate the risk of death.

  1. How is the influence of the length of period between diagnosed as having cancer and the death from heart related disease? From this study, this author found that only patients with acute leukemias, pancreatic cancer and nervous system malignancies did not have a seasonal pattern in heart diseases-related fatalities. However, those malignancies are known disease that is often diagnosed in advanced stage. Length of the life living with cancer should be taken into consideration.

Reply: We appreciate that comment. Therefore, we performed stratified analysis by early (within 1 year of diagnosis) versus late deaths (after 1 year since diagnosis). The seasonal pattern was virtually identical. This data was included in the original version of the manuscript in Table 2 and Figure 1.

  1. Geographical issue also might influence on present results. Living in city and living in countryside should have quite different in accessibility formedicalservice. Therefore, difference between living in city area and living in country side area might strong influence on present results. If only country side area shows seasonal difference on present results, accessibility for medical service but not season itself might be an important determinant factor on present results.

Reply: We appreciate that suggestion. We performed analysis stratified analysis per Metropolitan vs. Nonmetropolitan counties. The results are shown in Table 2 and Figure 1.

  1. Age also could influence on present results.

Reply: We appreciate that comment. Therefore, we performed stratified analysis younger patients vs. elderly (above 50 at diagnosis). The seasonal pattern was virtually identical. This data was included in the original version of the manuscript in Table 2 and Figure 1.

  1. In material and method section this author described as “Notably, this et of cases included only the entries of the first diagnosed primary malignancy. The month of death of each patient is not provided in the SEER database. Therefore, we estimated it for all patients based on the reported month and year of diagnosis and survival time in months. For the purposes of that estimation we accepted that the average duration of any month is 30.25 days.” However, I could not understand the meaning of those sentences. What type of report that showed month and year of diagnosis and survival time? Is this means SEER database have the information of month and year of diagnosis and survival time?

Reply: Your understanding is correct. SEER database provides information regarding the month and year of diagnosis, overall survival and cause of death per major entities.

  1. Severity of malignant disease at the time of diagnosed also should be taken into consideration.

Reply: You are right. Therefore we performed stratified analysis by main groups of malignancies. This is presented in the original manuscript in Table 2 and Figure 2.

  1. There are many cardiovascular risk factors that also regarded as risk factor for cancer. Therefore, many participants with cancer also might have a higher risk of cardiovascular risk factors. In this case, cardiovascular risk factor but not malignant disease determines the season of death from heart related disease. In this case, cancer itself never influence on the season of death from cardiovascular disease.

Reply: That is exactly our conclusion cancer does not influence the seasonal pattern of cardiac mortality which is probably driven by other factors independent of cancer diagnosis.

Reviewer 3 Report

You note at line 127: It was rational to accept that not all malignancies would have a seasonal variation in the mortality from heart diseases. Why? Cancer would not be expected to be protective. One could speculate that cancer patients have greater surveillance and earlier use of antibiotics might account for an observed reduction in the seasonable variability. That you did not see seasonality with acute leukemia, pancreatic cancer, and nervous system malignancies suggests, as you note in the discussion, might be due to a shorter duration of survival masking seasonal variation.

You note at line 157: “This question (whether cancer diagnosis usually associated with multimodal therapy that may have cardiac and vascular toxicity modifies the seasonal pattern of heart disease-related mortality) is of obvious biological and clinical importance. Again, this reviewer asks what is the basis for this statement? You address this to a very limited extent in your discussion, but it could be expanded to provide insight as to why this is of obvious importance beyond what is known about seasonal variations in the general population.

Author Response

You note at line 127: It was rational to accept that not all malignancies would have a seasonal variation in the mortality from heart diseases. Why? Cancer would not be expected to be protective. One could speculate that cancer patients have greater surveillance and earlier use of antibiotics might account for an observed reduction in the seasonable variability. That you did not see seasonality with acute leukemia, pancreatic cancer, and nervous system malignancies suggests, as you note in the discussion, might be due to a shorter duration of survival masking seasonal variation.

Reply: Different malignancies may differ in their aggressiveness and also type of therapy with differential cardiovascular toxicity. This idea is obviously supported by the lack of seasonality in more aggressive malignancies such as acute leukemias, pancreatic cancer and CNS tumors. As you noticed we discussed this in the discussion section. We have never claimed that cancer is protective rather it may “erase” any seasonal pattern because of the toxicity of previous anti-cancer therapy. Again this is a purely descriptive study and it cannot provide any ultimate mechanistic explanations. In our view any attempt to provide mechanistic explanations can be considered an unjustified over-interpretation and highly speculative.

You note at line 157: “This question (whether cancer diagnosis usually associated with multimodal therapy that may have cardiac and vascular toxicity modifies the seasonal pattern of heart disease-related mortality) is of obvious biological and clinical importance. Again, this reviewer asks what is the basis for this statement? You address this to a very limited extent in your discussion, but it could be expanded to provide insight as to why this is of obvious importance beyond what is known about seasonal variations in the general population.

Reply: We justified our statement as follows: This question is of obvious biological and clinical importance because the long-term cancer survivors (e.g. Hodgkin lymphoma patients) suffer from increased long-term mortality because of cardiac diseases as consequence of the cardiotoxic chemo- and radiotherapy at younger age.

Round 2

Reviewer 2 Report

 I have no further comment. I’m satisfied with this revised version of manuscript.

Reviewer 3 Report

I acknowledge the changes that have been made and the author's explanations regarding previous comments.